# Process Analysis on Milled Optical Surfaces in Hardened Stainless Steel

**Marvin Groeb * and Matthias Fritz**

Kern Microtechnik GmbH, 82438 Eschenlohe, Germany

* Correspondence: marvin.groeb@kern-microtechnik.com

**Abstract:** The capability to produce surfaces in optical quality is of rising prominence in the manufacturing industry. The die and mold industry have to meet rising requirements with regard to the surface finish and geometric precision, to keep pace with technological advances in sectors such as illumination, optical sensors, and fiber technology. This paper focuses on the challenges of developing a sample milling process for optical surface finishes in 53 HRC STAVAX stainless steel. Besides the expected process and tooling parameter variations, three prominent material analytics methods were used to evaluate all experiments. The tool wear was analyzed and monitored via SEM and EDS. To verify the milling process capabilities, a sample was produced through manual polishing and then analyzed for material faults via nanoindentation and BSE analysis. The finished results were measured for their surface roughness via bifocal laser microscopy and for their topography SEM. In the practical application, a surface roughness $R_a$ in the single digit nano-meter range was achieved. A stable finishing process was achieved at high rotational speed with feed rates up to 750 mm/min. A practical cleaning solution with the use of a mild lye was found.

**Keywords:** burnishing; milled optical surfaces; stavax; precision milling; UPM

## 1. Introduction

The capability to produce surfaces in optical quality is of rising prominence in the manufacturing industry. Sectors such as illumination, optical sensors, fiber technology, and also die and mold demand milled surfaces in the highest possible quality, whereas the need for automation and less manual production require a finished part right off the machine. The global injection molded plastic market has been valued at several hundred billion dollars in 2016 [1]. Consequently, research is being carried out into improving milled surfaces, with the arithmetical surface roughness Ra as the most often used quality parameter. The limiting factor of the achievable surface roughness is not only the machine, but a broader range of parameters, including but not limited to the process, tool, material, and cutting conditions [2]. Furthermore, nanoscale surfaces set rising requirements to the measuring and evaluation technique in quality control. The Society of the Plastics Industry (SPI) has set a standard for injection molding, where the highest achievable finish is grade A-1 with a typical surface roughness $R_a$ between 0.012 and 0.025 μm [3]. A surface roughness in the single digit nanometer range is desired. Traditionally, the injection mold is manufactured through several steps. First, the general shape of the part is roughed out on a milling machine, which is followed by a heat treatment to increase the durability of the mold. The desired shape is created through electrical discharge machining (EDM), which is an expensive and complex process requiring the creation of an electrode beforehand. Afterward, the form is polished manually in a labor-intensive process, which is largely dependent on the skill and experience of the worker. Shorter time to market, which is a general shortage of skilled employees as well as the advancing automation require a new approach. Contrary to the traditional use of EDM to create small features, the use of high-speed spindles enable productive application

of small tooling. Surface roughness also decreases with the use of HSM [4,5]. The desired surface quality is achievable through special coated tools or diamond cutting directly on the milling machine. Summarily, due to the limitations of the traditional production for injection molds, milling requires specialized tooling, a capable machine, and a fine control over the process.

## 2. Materials and Methods

In this paper, the process for creating high quality surfaces in hardened stainless steel via a (polycrystalline diamond) radius tool is explored. For this, the material is analyzed with a confocal laser microscope with regard to the surface roughness. A scanning electron microscope (SEM) is used to determine the topography, energy dispersive X-ray spectroscopy (EDS) for the elemental analysis of material anisotropies, and the wear of the tooling used. Nano indentation verifies the material isotropy as well as specimen hardness. Several specimens are produced under varying cutting conditions, which explores different forms of tooling.

The material used in all experiments is Stavax ESR (electro slag re-melting), produced by Uddeholm GmbH. Stavax ESR is a widely used stainless steel with high corrosion resistance and is well suited for micro molds. The composition [6] is 0.38% carbon, 0.9% silicon, 0.5% manganese, 13.6% chrome, and 0.3% vanadium besides the expected iron. The delivered specimens have a size of $30 \times 23 \times 102$ mm. The material is delivered in an annealed state and cut by saw. The smelter number of the batch used is A16002. Before hardening, the material is roughed out to create square and comparable specimen. A drafted surface with an inclination of 45° is roughed to enable 3 axis milling at a sensible point of engagement with the radius tooling used. The roughed-out material is, afterwards, cleaned with ethanol alcohol in an ultrasonic cleaner. Hardening is done with the Uddeholm proposed process. A hardness of 53 HRC ±1 HRC was achieved.

This paper focuses on the application of free-form milling regarding the surface finish. This finish is primarily dependent on the machine, cutting conditions, tool geometry, the environmental conditions, material property, chip formation, and vibration [2]. The surface topography is formed as a result of the tool-workpiece interaction. Therefore, the material removal mechanism is of greater importance [2]. Figure 1 shows the four dominant chip formation mechanisms in milling.

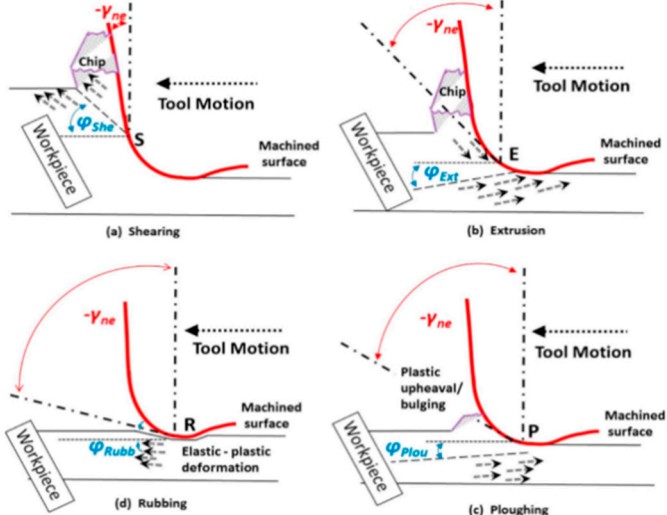

**Figure 1.** The four distinct chip formation mechanisms in milling [2].

While the shearing and extrusion process are dominant in normal machining, in precision machining (PM), the rubbing and ploughing mechanism (Figures 2.1 c and d) dominates [7]. Furthermore, the ratio of the undeformed chip thickness to the tool edge radius (relative tool sharpness (RTS)) is the defining parameter of the theoretically achievable surface roughness in milling processes [2,7].

In high-precision machining, the surface roughness can be in the same range as the uncut chip thickness [8]. The theoretically achievable surface roughness (see Figure 2) is a function of the tool radius ($r$) as well as the axial runout ($a$), which can be calculated using Equation (1) [9]. In this case, $f_z$ is the feed per tooth.

$$R_{th} = \frac{f_z^2}{8r} + a \qquad (1)$$

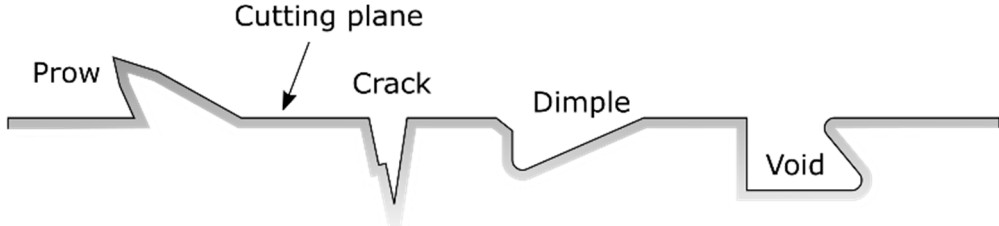

**Figure 2.** Kinematic roughness. $f_z$ is the feed per tooth. $R$ is the resulting roughness, whereas the resulting deviation from the ideal line is specified by e and a [9].

Moreover, a relative improvement to the surface can be achieved with the burnishing process [10], where the material is flattened under the tooling [11]. This process is seldom done on milling machines, and explored in this paper.

While the surface may have a good macroscopic appearance, small surface defects appear on the milled surface. These surface faults originate from the difference between microscale and macroscale cutting. This result from the so called 'size effect' means there is an increase in material shear flow stress as the chip thickness decreases. Because the fault size is so small, the SEM observation is necessary. Figure 3 shows the major surface faults along the cutting plane.

**Figure 3.** Surface faults in micro-milling.

In the previously used process by Kern, a polycrystalline diamond (PCD) coated radius ball mill was used with an ISO-strategy to finish the surface. The optical surface has a high visible gloss and a surface roughness $R_a = 0.015$ µm. Visible surface faults are grooves along the feed direction, which are generally very fine scratches in every orientation as well as an inconsistent gloss.

## 2.1. Bulk Material Analysis

In order to examine the bulk material used for anisotropies, a hardened sample is prepared with a hand polishing process. For this, the sample is first ground on a rotary grinding machine (Metko Gripo 2) with increasingly finer grit (from 80 to 4000 in 10 steps). The grinding direction on the sample is rotated for 90° before the next step and the grinding process is stopped once the grind marks from the previous step are completely removed. Water coolant is used during the grinding process. Following the last grinding step with 4000 grit sandpaper, the sample is cleaned and then polished on a cloth substrate with a diamond polishing compound with particles ranging from six micrometer down to a one micrometer diameter. An alcohol-based wetting fluid is used to reduce the friction. The polished sample is then analyzed for its topography in an SEM microscope, which is followed by nano-indentation.

## 2.2. Tool Wear Analysis

Analyzing the tool wear in precision machining is challenging. Because the surfaces are of a very high quality, very small tool wear effects or edge breakouts can have a huge impact. Moreover, a slightly worn or rounded edge is often not visible under optical microscopes. Therefore, this requires higher resolution imaging techniques. A well-established method is the use of an SEM microscope. The SEM is used to both judge wear in a qualitative way as well as through EDS to make an elemental analysis to detect build up edge phenomena.

## 2.3. Cleaning Study

Directly after milling, the samples are cleaned with compressed air. A black, tough residue is left on the sample. This film appears with refracting colors from different angles. In order to clean milled parts, a common approach is the use of ethanol in an ultrasonic bath. Since the residue could not be removed by ethanol, a series of organic solvents and bases were evaluated in their ability to remove the film. For this, a small drop of the solvent was placed on the sample. After 3 min, the drop was wiped away with a precision lab towel and the surface was optically examined.

## 2.4. Process Parameter Variation

In order to improve the surface finish, prolonging tool life and avoiding the previously mentioned surface faults, a number of process parameter variations were undertaken, which mainly adjusted the cutting speed, the feed rate, and the depth of cut. The samples were milled with an iso-distribution strategy (meaning the toolpath stepover is evenly distributed along the surface). Table 1 lists the used universal milling parameters, whereas Table 2 lists the spindle speed ($n$) variation and their resulting feed per rotation ($f_n$) as well as cutting speed ($v_c$) at 45°. Because the tool has no geometric set cutting edge, the more commonly used feed per tooth cannot be calculated.

**Table 1.** Universal milling parameters used in the experiments.

| Milling Parameter | Value |
| --- | --- |
| $a_p$ (Depth of Cut) | 5 μm |
| $a_e$ (width of cut) | 5 μm |
| Feed rate | 670 mm/min |
| Coolant | Flood coolant |
| Milling direction | Climb milling |
| Dynamics setting | Finishing |

**Table 2.** Cutting speed variation parameters.

| $n$ in k/min | 40 | 37 | 35 | 31 | 28 | 25 | 22 | 19 | 16 | 13 | 10 | 7 | 4 | 3 |
| --- | --- | --- | --- | --- | --- | --- | --- | --- | --- | --- | --- | --- | --- | --- |
| $f_n$ in μm | 19.0 | 20.5 | 22.3 | 24.5 | 27.1 | 30.4 | 34.5 | 40.0 | 47.5 | 58.4 | 76.6 | 109 | 190 | 253 |
| $v_c$ in m/min | 126 | 116 | 110 | 97 | 88 | 79 | 69 | 60 | 50 | 41 | 31 | 22 | 13 | 9 |

The feed rate is then gradually reduced from a value of 660 mm/min to 140 mm/min. Because the surface quality in all cutting processes is largely dependent on the force exerted between tool and material, the cutting depth was gradually reduced from 5 μm to 0.5 μm.

## 2.5. Tooling Type Variation

The used tooling in all experiments is a ball shaped radius milling tool (see Figure 4). The tool has a spherical tip with no geometric set cutting edge. During the majority of experiments, a polycrystalline diamond coated commercially available endmill (PCD-RB) was used. Different coatings were also researched. For this, a second polycrystalline diamond tool was made with an identical geometry, where the diamond surface was not laser edged after coating. A rougher film is visible. The tool looks

matte to the naked eye. Moreover, a tool was coated with a commercial hard coat (WAD-coating), which leads to a very even and smooth surface film. These tools were compared at identical milling parameters to the PCDR-RB used in the other experiments and the achieved surface finishes evaluated with the confocal laser scanning microscope. A feed rate of 500 mm/min at a rotational speed of 29,400 min$^{-1}$ was chosen. The stepover and depth of cut was 5 μm.

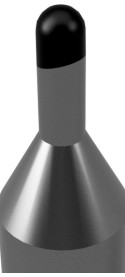

**Figure 4.** Appearance of the used ball radius tooling. The tip is coated.

## 3. Results

### 3.1. Bulk Material Anaylsis Results

The SEM analysis shows (see Figure 5a) a uniform surface mildly marred by scratches.

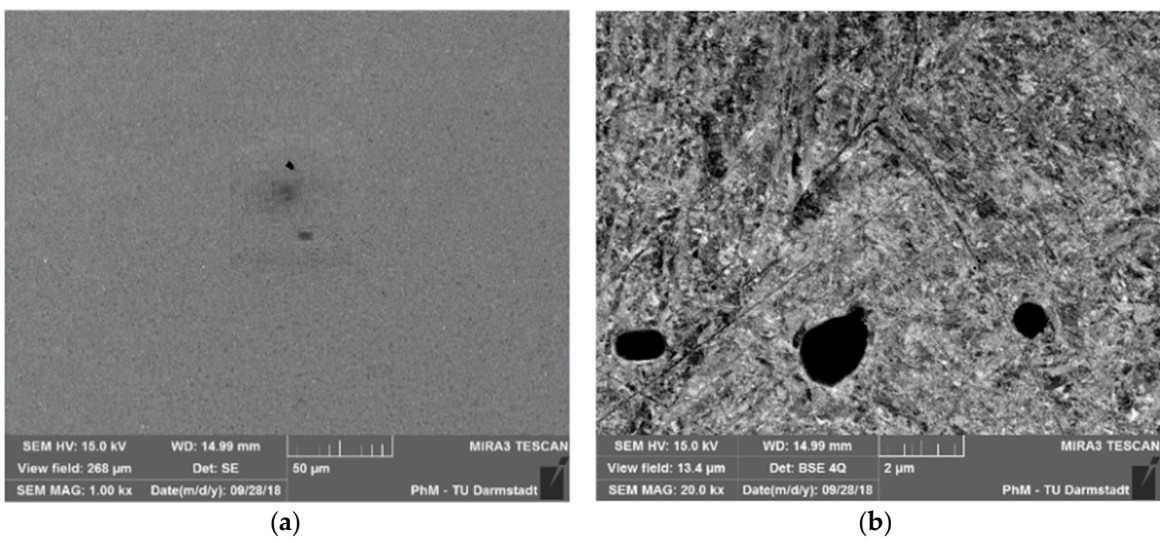

**Figure 5.** SEM (**a**) and BSE (**b**) image of the hand polished Stavax surface.

Small black dots appear all over the sample, for which the BSE imaging points to an accumulation of lighter elements.

In order to verify that the material is uniform in its behavior, a nano-indentation test was performed. Both the polished sample as well as an exemplary milled sample are glued to the specimen holder. A matrix of 10 by 10 indentations is created on every sample. An indentation depth of up to 550 nm was reached.

In Figure 6, it is clearly visible that the hand polished reference sample has a very uniform hardness of about 7.5 GPa. The milled reference sample shows a comparatively deeper hardness plateau, which reaches a hardness of up to 11 GPa. The standard deviation on the milled sample is 0.74 GPa, whereas the polished sample shows a standard deviation of 0.27 GPa. This leads to the conclusion of work hardening of the material during the milling process. A Vickers hardness test with a load of 0.5 kg showed a hardness for the milled sample of, on average, 670 HV 0.5 (58.9 HRC), which

further suggests work hardening of the sample. The polished sample exhibits a very uniform behavior, which makes vibrational surface faults arising from anisotropies in the material unlikely.

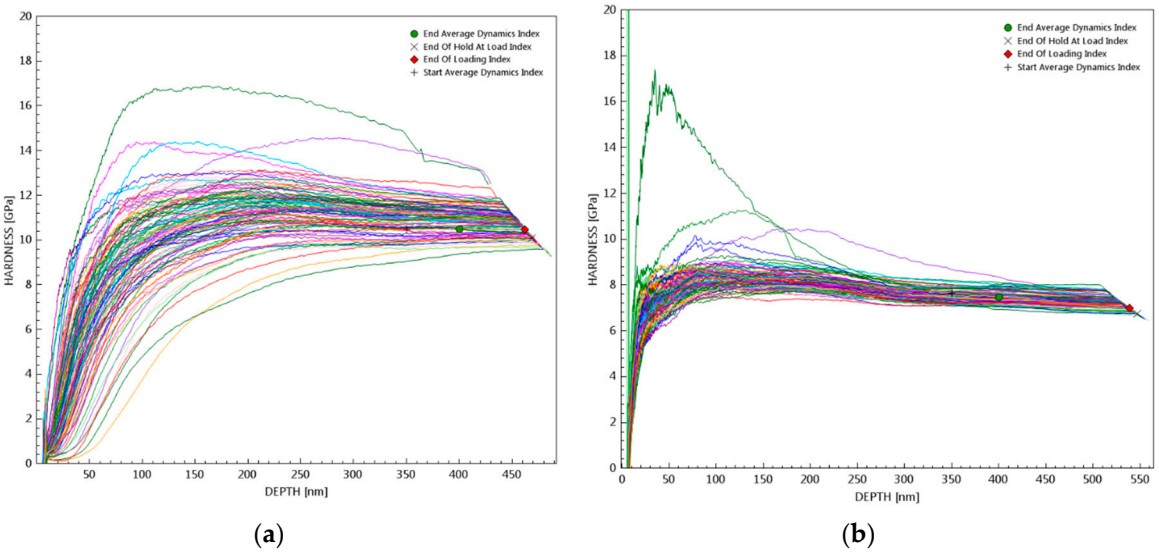

(**a**)　　　　　　　　　　　　　　　　　　　　　(**b**)

**Figure 6.** Nano indentation measurements of the polished (**a**) and milled (**b**) sample.

### 3.2. Tool Wear Analysis

Figure 7 highlights the areas of interest on the tool, whereas Figure 8 shows the recorded EDS spectra for the used tooling. Section 1 shows a lighter colored area on the tool. The elemental analysis shows mainly the used specimen material, which signifies adherence to the tool. In Sections 2 and 3, the thicker, unevenly formed particles are analyzed. Besides a carbon K alpha peak and several iron peaks, silicon as well as chrome are detected. Since these are the elements found in high concentrations in the used stainless steel, the particles on the surface appear to be a chemical adhesion of the work material, which is similar to a buildup edge. These particles stuck to the surface of the tool possibly explain the occurrence of scratches along the work direction. Conclusively, tool wear is clearly visible after a relatively short period of usage. The most likely wear process, in this case, is chemical adherence of the steel to the carbon of the diamond coating.

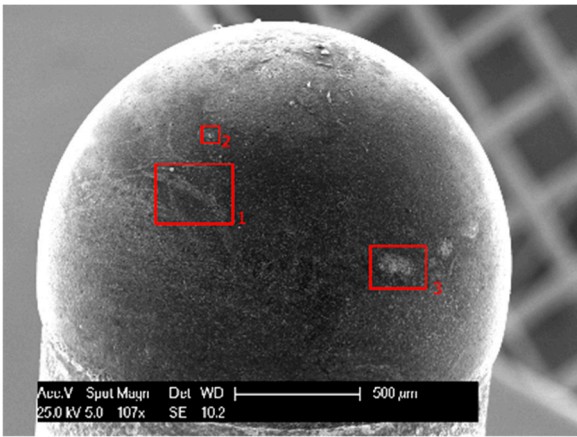

**Figure 7.** SEM overview of the used PCD-RB tooling.

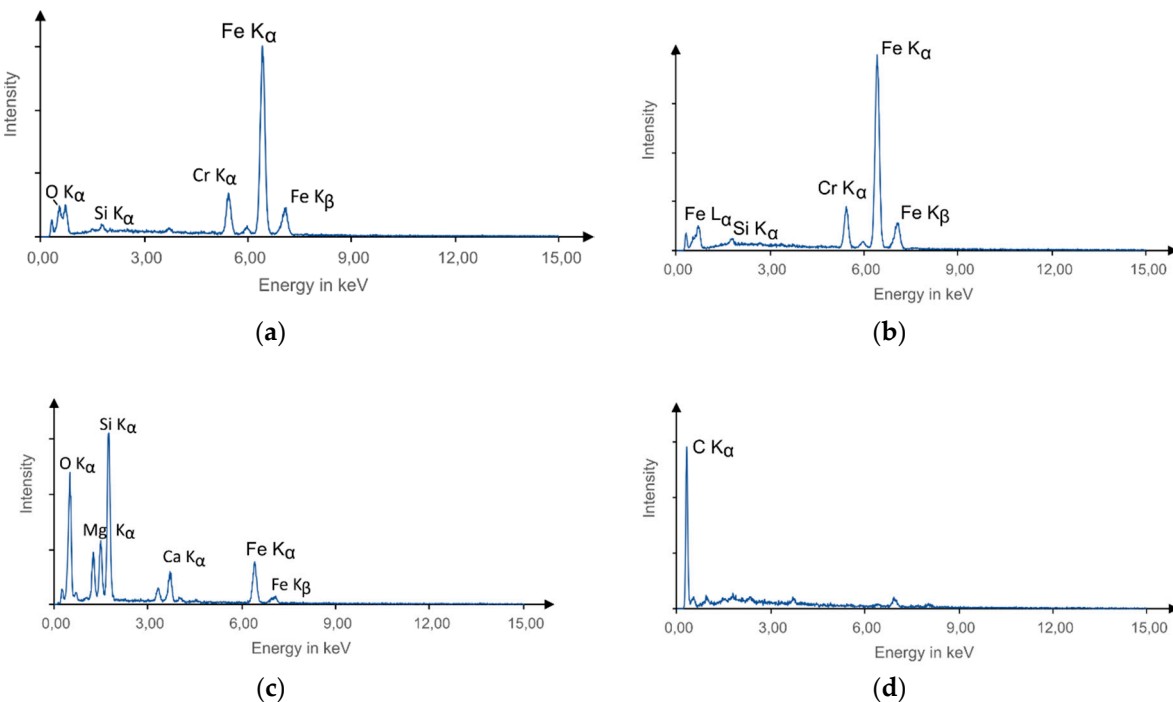

**Figure 8.** EDS spectra for the (**a**–**c**) selections 1–3 from Figure 7 as well as the undamaged coating (**d**).

### 3.3. Cleaning Study

Table 3 sums up the results. There, a "-" symbolizes no visible change, a "o" represents a small but ineffective change, and a "+" shows good film removal. It is clearly visible that the organic solvents were unable to remove the film, whereas the best result was obtained with NaOH lye. A concentration of 2M was deemed sufficient. After the application with lye, a cleaning best practice suggests a thorough washing in distilled water followed by spraying the sample with methanol and drying under a hot air fan. A residue free, optical surface was achieved.

**Table 3.** Overview of the different cleaning solvents used and their respective effectiveness.

| Chemical Substance | Cleaning Result |
| --- | --- |
| Ethanol alcohol | - |
| Isopropyl alcohol | - |
| Methanol alcohol | - |
| Acetone | - |
| Hexanol | - |
| Dichloromethane | o |
| 2M NaOH | + |
| 6M NaOH | + |

The surface roughness improved from $S_a$ 15 nm to $S_a$ 11 nm, which highlights the importance of the cleaning process of optical surfaces.

### 3.4. Process Parameter Variation

The surface shows a very high gloss uniform over the sample. To the naked eye, there is very little optical variance between the different spindle speed settings. The surface roughness was evaluated using a CLSM (Confocal laser scanning microscope, Confovis DuoVario). The $R_a$ value was chosen in conjunction with the $S_a$ instead of the $R_z$ value for several reasons. The $R_a$ value can, especially with contactless measurements, be tuned to represent a better surface quality than on average at the sample. For example, choosing the measurement path along smoother areas. The author picked a random path

for this paper, which was applied at the same position for every sample. Therefore, this avoids the statistical influence of the results. The $S_a$ value represents a more robust evaluation of high-quality surfaces and is, thus, a more fitting surface parameter. $R_z$ is not displayed in the following figures, since there is no identical S parameter equivalent. Moreover, the following figures are more concise and clear to the reader. The best surface finish with a $R_a$ of 9.5 nm was achieved at 28,000 $min^{-1}$. The surface finish stays below $R_a$ 20 nm for a very broad rotational speed band, which only increases significantly under 10,000 $min^{-1}$. The variance between the individual spindle speed settings is most likely caused by resonance. Figure 9 lists the surface roughness $R_a$ as well as $S_a$ for the experiment.

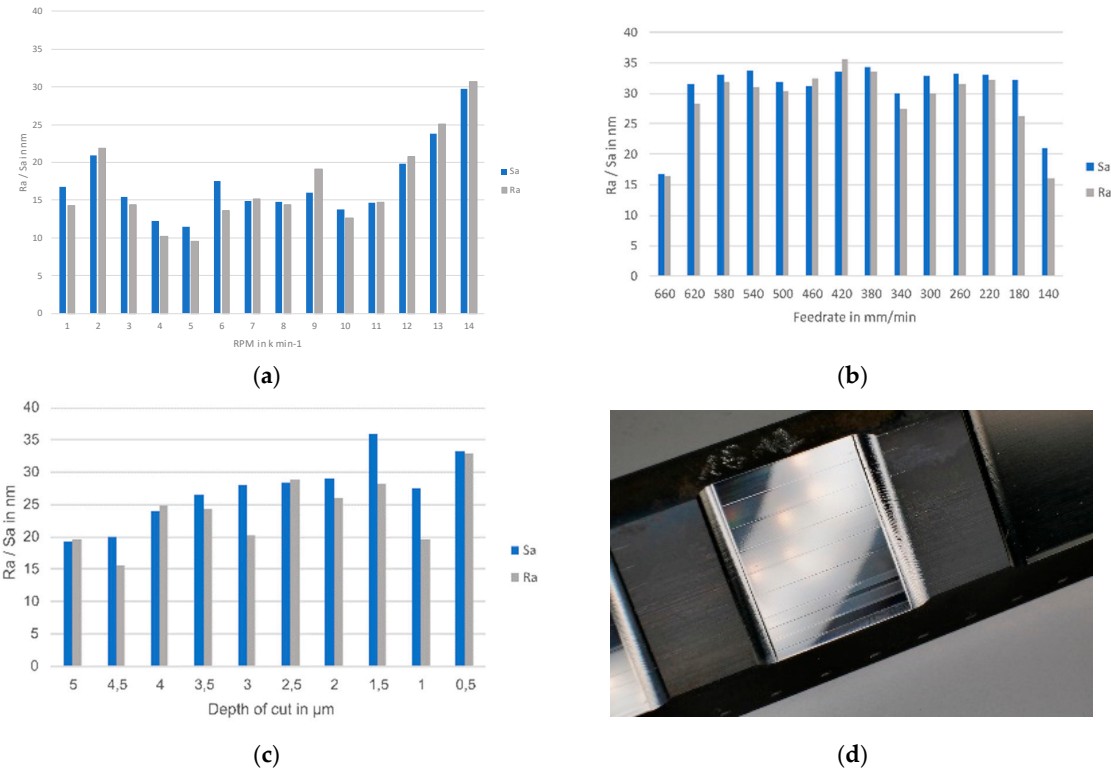

**Figure 9.** Surface roughness for the spindle speed variation (**a**), feed rate variation (**b**), and depth of cut variation (**c**). Picture of a finished specimen (**d**). The sample area division and gloss are easily recognizable.

The surface roughness does not significantly change with a decreasing feed rate (see Figure 9). The large change from 660 mm/min to 620 mm/min is most likely attributable to a break in the period of the PCD-RB milling tool. Figure 10 shows an exemplary confocal laser scanning microscope (CLSM) picture of the surface. Varying the feed rate had no discernible impact on the surface roughness. The sample surface under the CLSM appear almost identical, which is similar to an optical inspection.

It can be seen that, with a decreasing depth of cut, the surface roughness generally deteriorates (Figure 10). At the lower values, this is most likely because the pre-finishing operation is starting to show. The depth of cut has an influence on the gloss of the sample. With increasing DOC, a higher gloss was visible.

Figure 10 shows an SEM picture of the surface after the finishing pass. Regular grooves in the feed direction are visible. Comparing their length and period with the process settings as well as referencing the likelihood of a particle becoming adhered to the tooling (see Section 3.2), the groove surface fault mentioned in Section 2 is identifiable as a picked-up particle marring the surface.

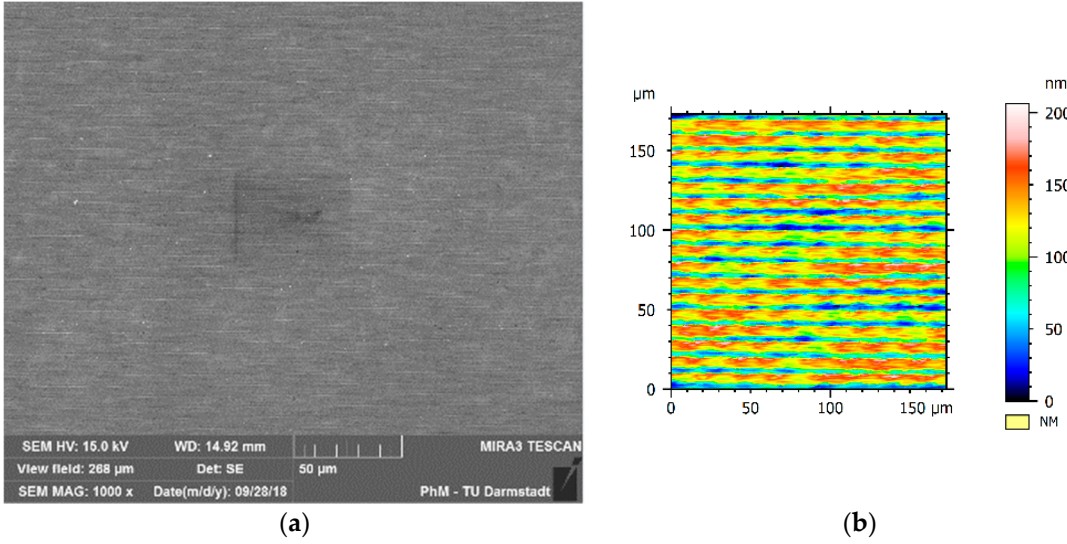

**Figure 10.** SEM (**a**) and CLSM (**b**) picture of the sample surface.

### 3.5. Tooling Variation

While the used PCD-RB tooling has proven to be an apt choice for the process, two different tools with identical shape have been used. The burnishing process in itself is more of a densification and coldworking of the surface. The influence of the tool smoothness is, therefore, of interest. Figure 11 shows the finished surface with the laser-smoothed PCD-RB on the left side, whereas the right side shows a solely coated endmill.

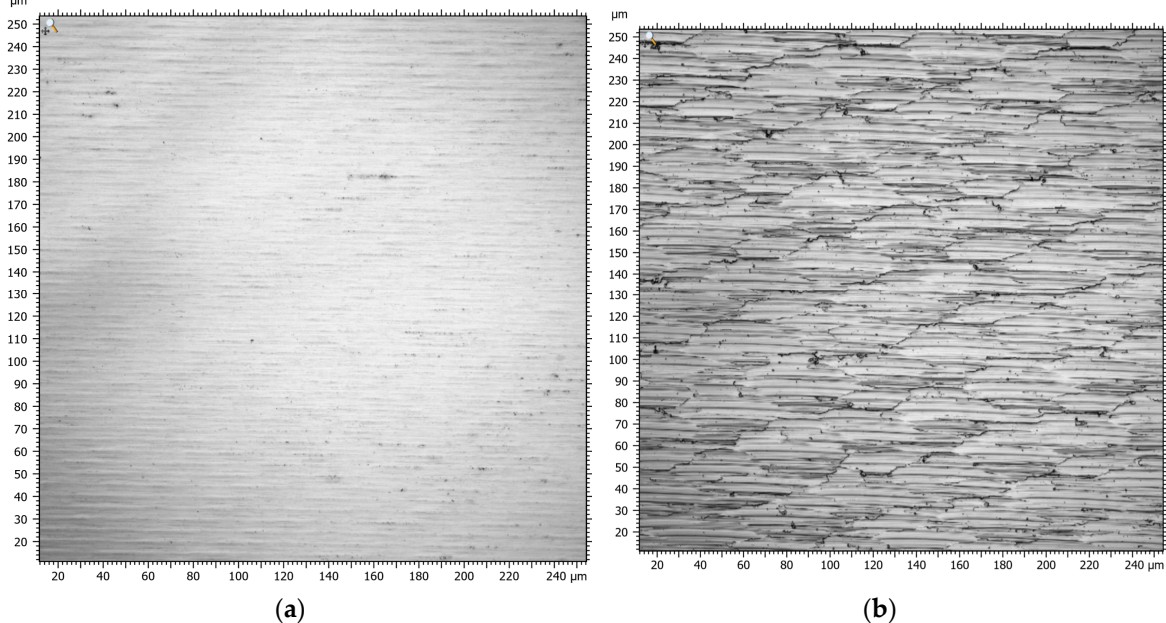

**Figure 11.** The finished surface with a laser-smoothed (**a**) and unsmoothed (**b**) PCD-RB tool.

It is clearly visible that the general appearance of the finished surface is still dominated by the pre-finishing operation. The smoothed tool shows a very fine texture, and a highly uniform surface. With the unsmoothed PCD-RB, a much more scale-like effect is visible. To the naked eye, the absence of a high gloss with the second tool is visible. Moreover, a tool failure after less than 10 h of finishing was recorded. The difference in surface roughness is also significant, which shows that $R_a = 0.014$ μm ($R_z = 0.077$ μm) on the laser-smoothed tool and $R_a = 0.127$ μm ($R_z = 0.566$ μm).

Since the used diamond coated tooling is quite expensive, and the burnishing process seems to improve with a smoother coated tool, a commercial hard-coating (WAD) was trialed with identical geometry. Figure 12 shows the surface produced by the hard-coated tool before and after tool failure. A higher tendency toward BUE (built-up edge)/adhesion was noticed, which further disturbs the surface. Before the tool failure, in some spots, a very smooth surface was produced. Measuring the surface roughness inside this area leads to $R_a = 0.063$ μm ($R_z = 0.415$ μm), whereas the surface after the failure shows $R_a = 0.116$ μm ($R_z = 1.04$ μm). The higher $R_z$ compared to the PCD-RB tools is related to the BUE phenomena. After only 3 mm$^2$ of milled area, the tool shows a failure, which is visible to the naked eye in rubbed coating.

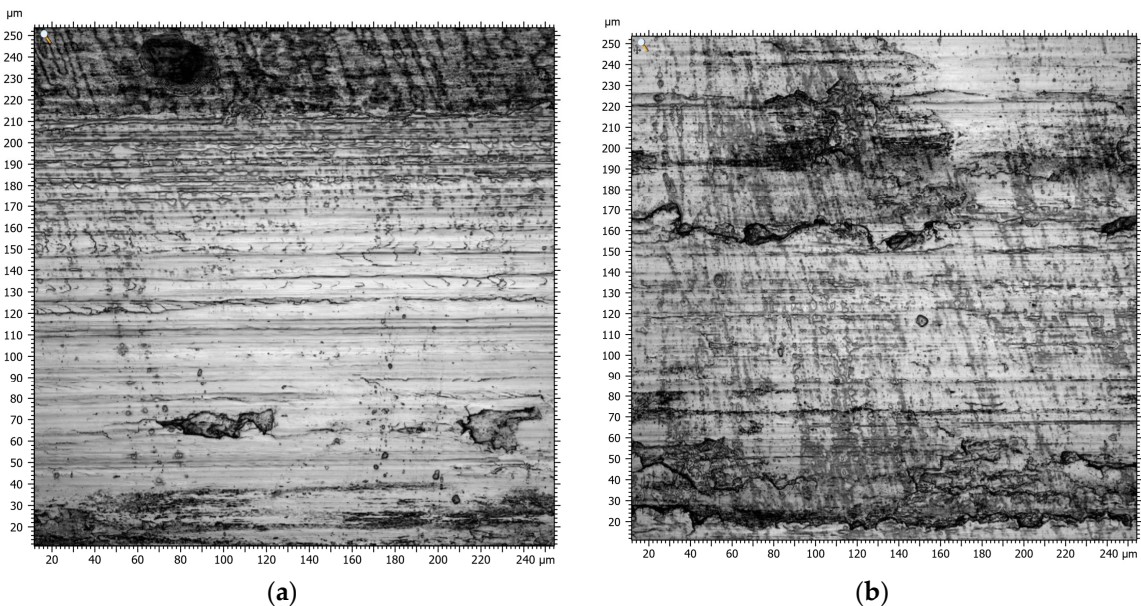

<center>(<b>a</b>)             (<b>b</b>)</center>

**Figure 12.** Surface produced with a hard-coated tool before (**a**) and after (**b**) tool failure.

## 4. Discussion

The stainless steel used in the experiments (Stavax by Uddeholm) was analyzed regarding its isotropy. First, a hardened specimen was manually polished. Indentation tests showed a hardness of 53 HRC, whereas the nanoindentation showed a steep increase in the surface hardness to a value of 7.5 GPa, which correlates to the normal indentation test. This correlates to the burnishing process [12]. SEM pictures showed a clean, uniform surface with good polishing behavior. BSE analysis suggests larger crystallographic grains and roughly one micrometer-sized spherical inclusions of a lighter element, which is most likely carbon. This indicates a very uniform material after hardening. The surface faults visible in the milled sample specimen are, therefore, not related to the material, which proves Stavax is an apt choice for the finishing process proposed in this paper.

A subjectively good surface quality is defined by the gloss exhibited by the sample. It can be marred by the surface faults apparent, and the surface roughness is an easily measurable quality. Nevertheless, the challenge to combine these three factors into one process is difficult to achieve.

Spindle speed, feed, depth of cut, and their influence on the surface roughness were evaluated. The spindle speed has no noticeable impact on the surface roughness. This is contrary to a milling process with a geometric set cutting edge [13]. The surface roughness improves with a higher DOC, and a slower feed rate returns a better surface as well. A higher depth of cut also increases the gloss of the sample surface. A tool analysis in combination with the process variation showed that the grooves marring the surface are caused by small particles adhering to the tool. At higher rotational speeds, the occurrence decreased.

Because the polycrystalline diamond tool left a black, tough residue on the surface, the possibility of chemically cleaning the sample was considered. A 2M solution of NaOH lye proved to be very effective in cleaning the film. Washing the sample in distilled water as well as finishing the surface with methanol alcohol proved to be a working and cost-effective solution for cleaning the surface. The surface roughness measurement was improved from 15 nanometers to 11 nanometers by the used approach.

A tooling variation from the laser finished PCDRB to a non-laser finished PCDRB and a WAD type tool was undertaken. While, in principle, the burnishing process can be done with any tool of the same geometry, it has been shown that a smooth coat with high durability is required to make a functioning tool. So far, for steel applications, this seems best matched by the laser smoothed PCD-RB.

## 5. Conclusions

The process analysis on the burnishing tool used in this paper has shown to be an apt choice in improving the surface to the highest requirements. Besides achieving a surface finish, $R_a$ in the single-digit nanometer range, improvements to the mechanical characteristics like hardness were achieved. A practical cleaning solution was found utilizing a weak lye as solvent. A stable cutting process, at a wide set of usable process parameters, was found, with optimal settings at a rotational speed of 28,000 $min^{-1}$ and a feed rate of 670 mm/min.

This paper has only evaluated one specific type of stainless steel. In mold making, besides ESR steels (of which Stavax is a good representative), powder metallurgic steels such as Elmax are very common as well. Because of the finer grain size, their polishing performance is often higher, which enables lower surface roughness. Moreover, a high surface finish in titanium, aluminum, and other metals is sought after, which suggests a material study with the existing process.

**Author Contributions:** Conceptualization, M.G. and M.F.; methodology, M.G.; formal analysis, M.G.; investigation, M.G.; resources, M.F.; data curation, M.G.; writing—original draft preparation, M.G.; writing—review and editing, M.G.; visualization, M.G.; supervision, M.G.; project administration, M.G.; funding acquisition, M.F.

**Funding:** This research received no external funding.

**Conflicts of Interest:** The authors declare no conflict of interest.

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
