# Peer review of "Process Analysis on Milled Optical Surfaces in Hardened Stainless Steel"

_jmmp, doi:10.3390/jmmp3030067_

Round 1

Reviewer 1 Report

The article titled "Process analysis on milled optical surfaces in hardened stainless steel" is prepared for publication in an interesting way in the Journal of Manufacturing and Materials Processing.

However, some disadvantages can be found.

Which needs to be improved:

1. The article ends with a discussion. This is an important chapter, but it is required to put at the  end of the publication a summary and conclusions resulting from research or consideration.

2. In line 84 there is in the formula (1) the designation "Fz". In contrast, in Figure 2, in line 86 is given "fz". Is this the same physical quantity? It seems to me that the letter designations should be standardized throughout the whole study. If the designation is lowercase "fz" then this should be found throughout the article.

3. In table 1 (line 144), the depth and width of cut are indicated by big letters, it should be lowercase "ap" and "ae".

4. Figure 6 (line 184) is very poorly readable. It's worth changing descriptions and characteristics.

A very large number of curves blurs the image, and there is no way to read the results.

After making corrections, you can publish.
  Accept after minor revision.

Author Response

1. The article ends with a discussion. This is an important chapter, but it is required to put at the  end of the publication a summary and conclusions resulting from research or consideration.

A conclusion has been added, moreover larger portions of the paper reordered to fit the required changes by the other reviewers.

2. In line 84 there is in the formula (1) the designation "Fz". In contrast, in Figure 2, in line 86 is given "fz". Is this the same physical quantity? It seems to me that the letter designations should be standardized throughout the whole study. If the designation is lowercase "fz" then this should be found throughout the article.

It has been changed to lowercase fz.

3. In table 1 (line 144), the depth and width of cut are indicated by big letters, it should be lowercase "ap" and "ae".

Corrected!

4. Figure 6 (line 184) is very poorly readable. It's worth changing descriptions and characteristics. 

A very large number of curves blurs the image, and there is no way to read the results.

The data came from an external source, and no individual data was given. I have increased size and resolution of the picture, moreover the text below the picture sums up the majority of findings.

Reviewer 2 Report

The conducted studies are undoubtedly interesting, but the structure of the article requires substantial improvement.

According to the reviewer, the Abstract contains an unnecessary introduction that is worth cutting out. The meaning of the Abstract - in the formulation of the task of the study, descriptions of key techniques and conclusions. It may be worth starting with the phrase "This paper focuses on .." and describe the results in more detail.

The keywords are presented in too general form. It may be worth specifying the area of research (for example, "milled optical surfaces", etc.)

Introduction, as a rule, involves a review of existing research in this area, showing the level of development of science and justifying the need and relevance of further research ("There are a number of studies, but the question ... has not been studied enough, therefore the aim of the work is ...") . The reviewer considers it inadequate to do a review of the state of science, relying on just three papers (two review articles and a standard). Such an introduction is more consistent with the advertising, rather than scientific article. The introduction should be substantially supplemented and expanded.

Text starting at line 49 should be moved to the next section ("Materials and Methods").

What is the scientific novelty of this article? A standard process, standard (commercially available) cutting tools, etc. are used. The reviewer recommends adding more focused conclusions. Usually used special section "Conclusions".

Figure 4. Presents not the geometry, but the appearance of the tool. Geometry usually involves specifying dimensions, angles, etc.

Figures 5a and 10a should be magnified. In the present form it is difficult to understand anything.

The text in Figures 6 and 8 should be in a higher resolution - now it is practically not readable.

Figure 7. The designation of areas 1,2,3 is practically unreadable - you need to significantly increase the font size.

These areas (1,2,3) must also be shown with a significantly higher magnification.

Figure 9 - it is better to designate as a, b, c, d

Figure 11 - it is also worth using the notation (a) (b)

What is WAD-coating?

The question of the presence of a built-up edge is always quite complicated and controversial. Perhaps, in this case, it is simply an adherent.

Authors need to eliminate typos, especially those associated with upper case (kmin-1; mm ^ 2)

What is the nature of "tool failure"? It is also advisable to give an image of this "failure", not just a short description.

Why are these two coatings compared (WAD and PCD-RB)? Why were these coatings selected?

Author Response

The conducted studies are undoubtedly interesting, but the structure of the article requires substantial improvement.

According to the reviewer, the Abstract contains an unnecessary introduction that is worth cutting out. The meaning of the Abstract - in the formulation of the task of the study, descriptions of key techniques and conclusions. It may be worth starting with the phrase "This paper focuses on .." and describe the results in more detail.

The abstract was shortened and results in higher detail added.

The keywords are presented in too general form. It may be worth specifying the area of research (for example, "milled optical surfaces", etc.)

This is a very valid point and has been adressed.

Introduction, as a rule, involves a review of existing research in this area, showing the level of development of science and justifying the need and relevance of further research ("There are a number of studies, but the question ... has not been studied enough, therefore the aim of the work is ...") . The reviewer considers it inadequate to do a review of the state of science, relying on just three papers (two review articles and a standard). Such an introduction is more consistent with the advertising, rather than scientific article. The introduction should be substantially supplemented and expanded.

The introduction was supplemented with more literature sources and especially a short overview of burnishing techniques.

Text starting at line 49 should be moved to the next section ("Materials and Methods").

Followed this advice.

What is the scientific novelty of this article? A standard process, standard (commercially available) cutting tools, etc. are used. The reviewer recommends adding more focused conclusions. Usually used special section "Conclusions".

A conclusion has been added, and portions of the methods and discussion rewritten.

Figure 4. Presents not the geometry, but the appearance of the tool. Geometry usually involves specifying dimensions, angles, etc.

Figures 5a and 10a should be magnified. In the present form it is difficult to understand anything.

The text in Figures 6 and 8 should be in a higher resolution - now it is practically not readable.

Figure 7. The designation of areas 1,2,3 is practically unreadable - you need to significantly increase the font size.

These areas (1,2,3) must also be shown with a significantly higher magnification.

Figure 9 - it is better to designate as a, b, c, d

Figure 11 - it is also worth using the notation (a) (b)

All figures have been adjusted in size and resolution.

What is WAD-coating?

WAD-coating is the commercial hardcoat used by the company Zecha. Naturally, they do not want to publish in any way what this coating is made off. I have changed the paragraphs to "commercial grade hard coating"

Authors need to eliminate typos, especially those associated with upper case (kmin-1; mm ^ 2)

I have rechecked and am hopeful to have found all typos.

Why are these two coatings compared (WAD and PCD-RB)? Why were these coatings selected?

A section adressing this has been added.

Reviewer 3 Report

The authors report a study on the milling process of hardened stainless steel for optical surfaces.

This research paper is relevant, well founded and discussed, and of interest to the audience of this journal. After the analysis of the content and style of the manuscript, there are some important observations. Under this considerations, major changes are recommended. This manuscript can be considered for publication after addressing the following comments:

1.     Abstract needs to address some values regarding your main findings as well.

2.     Some redaction mistakes have been detected, please take a moment to review and correct that.

3.     Although Ra may be the most common parameter in the roughness evaluation (L33), for the study of machining parameters behavior, others roughness parameters like Rz of Rt, can provide more information than the average roughness.

4.      Literature background needs to be completed the manuscript is based on not enough references to corroborate the study, for this purpose, the addition of some references is recommended. Please read and consider to use the following works in your manuscript as well:

-        Materials 201912(6), 860; https://doi.org/10.3390/ma12060860

-        Appl. Sci. 20199(5), 842; https://doi.org/10.3390/app9050842

-        Materials 201912(12), 2015; https://doi.org/10.3390/ma12122015

-        https://doi.org/10.1016/j.measurement.2018.07.058

-        https://doi.org/10.1016/j.promfg.2017.07.104

-        https://doi.org/10.1016/j.procir.2018.05.032

-        https://doi.org/10.1016/j.proeng.2015.01.491

-        https://doi.org/10.1016/j.wear.2019.02.018

-        https://doi.org/10.1016/j.procir.2018.05.076

5.      A deeper description of the Stavax ESR nedds to be included in the manuscript. In this sense, the information about composition, main properties and a justification of the use of this material is recommended.

6.      Call to references in the manuscript needs to be ordered. Can be observed that the reference [7] appears following to the reference [3] L76.

7.      Sections and subsections of the paper are not structured correctly, in the methodology section appears content that must be transferred to other sections. In the methodology section should only appear a description of the methods and techniques that have been used in the experimental phase.

8.      Roughness evaluation is one of the most important topic of the manuscript, however, the roughness measurement device used to evaluate the surface finish of the machined parts have not been described.

9.      Figure 4 needs to address a scale bar.

10.   Please review the figure titles.

11.   Figure 5 needs to be described correctly.

12.   Colors used to define Sa and Ra in figure 9 are different for the first chart and may be confused, please use homogeneous colors for similar analysis.

13.   Figures needs to be separated using a,b,c,d,   etc.

14.   Conclusion section, entitled discussion, needs to be revised and main findings needs to be addressed.

Author Response

1.     Abstract needs to address some values regarding your main findings as well.

Abstract has been largely rewritten, main findings added.

2.     Some redaction mistakes have been detected, please take a moment to review and correct that.

Adressed this.

3.     Although Ra may be the most common parameter in the roughness evaluation (L33), for the study of machining parameters behavior, others roughness parameters like Rz of Rt, can provide more information than the average roughness.

4.      Literature background needs to be completed the manuscript is based on not enough references to corroborate the study, for this purpose, the addition of some references is recommended. Please read and consider to use the following works in your manuscript as well:

 More literature background, especially in regards to the burnishing process have been added along with two of your suggested papers. Nevertheless, the rest of them  was a) unkown to me and b) made for an interesting read - thanks!

5.      A deeper description of the Stavax ESR nedds to be included in the manuscript. In this sense, the information about composition, main properties and a justification of the use of this material is recommended.

The chemical composition and industry usage has been added.

6.      Call to references in the manuscript needs to be ordered. Can be observed that the reference [7] appears following to the reference [3] L76.

Adressed.

7.      Sections and subsections of the paper are not structured correctly, in the methodology section appears content that must be transferred to other sections. In the methodology section should only appear a description of the methods and techniques that have been used in the experimental phase.

Adressed. Large portions restructured, in compliance with the other reviewers.

8.      Roughness evaluation is one of the most important topic of the manuscript, however, the roughness measurement device used to evaluate the surface finish of the machined parts have not been described.

The confocal laser scanning microscope type (by confovis) has been added.

9.      Figure 4 needs to address a scale bar.

Rewritten Figure title, as reviewers pointed out it is only the appearance, not the geometry of the tool. A scale bar has not been added as it's a ISO view of the tool. The use of this figure is to further support the description that the tool has no geometric set cutting edge.

10.   Please review the figure titles.

Addressed.

11.   Figure 5 needs to be described correctly.

Addressed.

12.   Colors used to define Sa and Ra in figure 9 are different for the first chart and may be confused, please use homogeneous colors for similar analysis.

Colors are now identical.

13.   Figures needs to be separated using a,b,c,d,   etc.

Addressed.

14.   Conclusion section, entitled discussion, needs to be revised and main findings needs to be addressed.

Section conclusion added as well as parts of discussion rewritten.

Round 2

Reviewer 2 Report

Since the authors took into account the recommendations of the reviewer and were able to improve the quality of the manuscript, the reviewer believes that the existing manuscript can be recommended for publication.

Author Response

Thank you for your comment. We have included one further literature source ([13]) and clarified the use of the Parameters Ra and Sa (line 247ff) in request to reviewer #3.

Reviewer 3 Report

Some of the considerations have been adressed on the manuscript. The initial manucript was improved properly, however, there are some previous suggestions that need to be considered before the publication of this manuscript.

 1. Although Ra may be the most common parameter in the roughness evaluation (L33), for the study of machining parameters behavior, others roughness parameters like Rz of Rt, can provide more information than the average roughness. Could you justify the use of the Ra parameter instead others?

2. Literature background needs to be completed the manuscript is based on not enough references to corroborate the study, for this purpose, the addition of some references is recommended. also, if the references are "unknown for you", may offers new information about relevant works related with the topic of your research. We think that the suggested references may improve the manuscript properly, for that, we encourage to read and consider to use the following works in your manuscript as well:

-        Materials 201912(6), 860; https://doi.org/10.3390/ma12060860

-        Appl. Sci. 20199(5), 842; https://doi.org/10.3390/app9050842

-        Materials 201912(12), 2015; https://doi.org/10.3390/ma12122015

-        https://doi.org/10.1016/j.measurement.2018.07.058

-        https://doi.org/10.1016/j.promfg.2017.07.104

-        https://doi.org/10.1016/j.procir.2018.05.032

-        https://doi.org/10.1016/j.proeng.2015.01.491

-        https://doi.org/10.1016/j.wear.2019.02.018

-        https://doi.org/10.1016/j.procir.2018.05.076

Author Response

Dear Reviewer,

Regarding point 1:

I have included a paragraph on line 247ff specifying the use of Ra and Sa instead of Ra and Rz. I hope this adquately answers your request for clarification.

Regarding the literature sources:

https://doi.org/10.3390/ma12060860

I have included this source ([5])

https://doi.org/10.3390/app9050842

I have not found this paper applicable, as we do not use a face milling process. Moreover, we don't consider milling dynamics in our paper.

https://doi.org/10.3390/ma12122015

While an exploration the the chemical adhesion processes would be of interest to the wear mechanism in our paper, we believe the adhesion chemistry of titanium is far different from our material (stainless steel), thus the paper not applicable.

https://doi.org/10.1016/j.measurement.2018.07.058

We have not found a relation to our paper, as we do not use a ball mill cutter (no geometric set cutting edge) as well as a "stub" tool, where the deflection is far from the (more than 1 order of magnitude longer!)tools used in the suggested paper.

https://doi.org/10.1016/j.promfg.2017.07.104

Again, the material and process does not apply to our subject.

 https://doi.org/10.1016/j.procir.2018.05.032

As we do not create a parameter model, and do not use a tool with a geometric set cutting edge, we again found no application for this paper.

https://doi.org/10.1016/j.proeng.2015.01.491

We have included this paper ([4)]

https://doi.org/10.1016/j.wear.2019.02.018

Again, the material and process does not apply to our subject.

https://doi.org/10.1016/j.procir.2018.05.076

We have included this paper. ([13])